# Differential Protein Expression of *Taenia crassiceps* ORF Strain in the Murine Cysticercosis Model Using Resistant (C57BL/6) Mice

**DOI:** 10.3390/pathogens12050678

**Published:** 2023-05-04

**Authors:** Lucía Jiménez, Mariana Díaz-Zaragoza, Magdalena Hernández, Luz Navarro, Ricardo Hernández-Ávila, Sergio Encarnación-Guevara, Pedro Ostoa-Saloma, Abraham Landa

**Affiliations:** 1Departamento de Microbiología y Parasitología, Facultad de Medicina, Universidad Nacional Autónoma de México, Ciudad Universitaria, A.P. 70228, Mexico City 04510, Mexico; lucia.jimenez@facmed.unam.mx; 2Departamento de Ciencias de la Salud, Centro Universitario de los Valles, Universidad de Guadalajara, Carretera Guadalajara-Ameca Km. 45.5, Guadalajara 46600, Mexico; 3Centro de Ciencias Genómicas, Universidad Nacional Autónoma de México, Av. Universidad 565, Chamilpa, Cuernavaca 62210, Mexico; 4Departamento de Fisiología, Facultad de Medicina, Universidad Nacional Autónoma de México, Ciudad Universitaria, A.P. 70228, Mexico City 04510, Mexico; 5Departamento de Inmunología, Instituto de Investigaciones Biomédicas, Universidad Nacional Autónoma de México, Ciudad Universitaria, A.P. 70228, Mexico City 04510, Mexico

**Keywords:** *Taenia crassiceps*, cysticerci, Th1 response, C57BL/6 mouse, proteomics

## Abstract

A cysticercosis model of *Taenia crassiceps* ORF strain in susceptible BALB/c mice revealed a Th2 response after 4 weeks, allowing for the growth of the parasite, whereas resistant C57BL/6 mice developed a sustained Th1 response, limiting parasitic growth. However, little is known about how cysticerci respond to an immunological environment in resistant mice. Here, we show that the Th1 response, during infection in resistant C57BL/6 mice, lasted up to 8 weeks and kept parasitemia low. Proteomics analysis of parasites during this Th1 environment showed an average of 128 expressed proteins; we chose 15 proteins whose differential expression varied between 70 and 100%. A total of 11 proteins were identified that formed a group whose expression increased at 4 weeks and decreased at 8 weeks, and another group with proteins whose expression was high at 2 weeks and decreased at 8 weeks. These identified proteins participate in tissue repair, immunoregulation and parasite establishment. This suggests that *T. crassiceps* cysticerci in mice resistant under the Th1 environment express proteins that control damage and help to establish a parasite in the host. These proteins could be targets for drugs or vaccine development.

## 1. Introduction

*Taenia solium* seriously affects human health in many Latin American, Asian, and African countries. It causes neurocysticercosis, which is the principal causes of epilepsy associated with infectious disease worldwide [1].

Because of the close phylogenetic relationship between *Taenia crassiceps* and *T. solium* and their antigenic and genomic similarity, the murine cysticercosis model for *T. crassiceps* has been widely used to study several physiological aspects of cysticercosis [2,3,4]. The resistance and susceptibility to *T. crassiceps* infections have been associated with the endocrine, genetic, and immune response in mice [5,6,7]. The susceptible BALB/c mouse favors a Th2 response and parasites’ growth with high levels of interleukin 4 (IL-4), IL-10 IL-13, IL-9, and transforming growth factor-beta (TGF-β), whereas the resistant C57BL/6 mouse develops a predominant Th1 response that limits the parasitic growth [8,9,10,11].

On the other hand, the immune response is influenced by parasite molecules. Cysticerci secretion/excretion (E/S) products promote a Th2 response with alternatively activated macrophages and T regulatory (Treg) cells in BALB/c mice [7,12,13,14]. Likewise, other parasite molecules drive toward the Th1 response with high levels of gamma interferon (IFN-γ) and IL-2, such as the cysticerci peptides KETc-1, KETc-2, and the enzyme glutathione transferases (SGSTF) [15,16]. In addition, antioxidant enzymes such as Cu,Zn superoxide dismutase, thioredoxin1, 2-Cys peroxiredoxin, and Glutathione transferases from Taenia participate in the protection and regulation of immune response. These antigens and enzymes have been proposed as vaccine candidates [17,18]. However, little is known about the protein expression when cysticerci are in a resistant host with a Th1 environment.

In the present study, we used 2-dimensional electrophoresis (2DE) and MALDI-TOF-mass spectrometry to determine the protein expression profile of cysticerci during an infection of C57BL/6 mice in a Th1 environment. In addition, we identified those proteins involved in repair, establishment, and defensive roles.

## 2. Materials and Methods

### 2.1. Mice

Female mice C57BL/6 and BALB/c strains were kept in the animal facilities at the Instituto de Investigaciones Biomedicas, UNAM, under controlled conditions of temperature (22 °C), a pathogen-free environment, relative humidity of 50–60%, 12-h dark-light cycles, and free access to food and water. The Institutional Ethics Committee approved all animal protocols (permission no. 2015–175).

### 2.2. Resistant Mice Infection with the T. crassiceps Cysticerci ORF Strain

C57BL/6 mice were intraperitoneally infected with ten cysticerci of 2–3 mm in 100 µL of sterile PBS; cysticerci were obtained from a female BALB/c with 6 months of infection. Infected mice were euthanized at 2 weeks, 4 weeks, and 8 weeks of infection in a CO_2_ chamber. The cysticerci were extracted from the peritoneal cavity, counted, washed 3-times with PBS in the presence of a protease inhibitor (Halt™ Protease Inhibitor Cocktail, Thermo Scientific, Waltham, MA, USA), and stored at −70 °C until used.

### 2.3. Spleen Cells Isolation and Cytokine Detection

Spleens from infected or non-infected mice were surgically removed after euthanasia under sterile conditions and splenocytes were isolated as previously reported [16]. Spleen cells (2 × 10^6^/mL) of infected and non-infected mice were stimulated with 5 µg/mL of *T. crassiceps* crude extract and cultured during 72 h at 37 °C and 5% CO_2_. Tumor necrosis factor (TNF), IL-12, IFN-γ, IL-5, IL-6, and IL-4 were quantified in the culture supernatants by ELISA, according to the manufacturer’s instructions (Mini ELISA Development Kit PEPROTECH). Assays were performed in triplicate.

### 2.4. Two-Dimensional Gel Electrophoresis (2DE)

Protein extract was prepared with the frozen cysticerci, as previously reported [19], in 2DE buffer (8 M urea, 50 mM DTT, 2% CHAPS, 2% Ampholine^®^, pH 3–10) (Bio-Rad, Hercules, CA, USA) and protease inhibitor (Halt™ Protease Inhibitor Cocktail). The protein concentration was measured using the Bradford method [20]. A total of 100 µg of protein from each sample was applied to IPG strips (GE HealthCare, Chicago, IL, USA) in an immobilized pH gradient from 3 to 10 by isoelectric focusing at 10,000 V over 8 h. Next, the strips were equilibrated with one wash for 10 min with 8 M urea, 0.375 M Tris, pH 8.8, 2% SDS, 20% glycerol, and 2% (*w*/*v*) DTT, and another wash for 10 min with 8 M urea, 0.375 M Tris pH 8.8, 2% SDS, 20% glycerol, and 2.5% (*w*/*v*) iodoacetamide. The proteins were separated in a second dimension by SDS-PAGE in a buffer of 25 mM Tris, pH 8.3, 250 mM glycine, and 0.01% SDS.

### 2.5. Proteomics Analysis

Next, 2DE gels were digitized on an HP Scanjet-G4050 scanner with a resolution of 300 DPI and analyzed using PDQuest™ 2DE Software (Bio-Rad) to determine expression levels. Master images were created from 3 repetitions of each 2DE image from 2-, 4-, and 8-weeks infection time of 2 separate experiments. The coordinates of each spot were calculated according to isoelectric point markers from 2DE standards (Bio-Rad). Data were normalized with triosephosphate isomerase enzyme (TPI), which depicted a constant level of expression during infection.

### 2.6. Statistical Analysis

One or two-way analysis of variance (ANOVA) with Tukey’s multiple comparison test was used for the statistical analyses of collected data, and *p* values < 0.05 were considered significant.

### 2.7. MALDI-TOF Mass Spectrometry Protein-Identification

Proteins with up- or down-expression levels between 70 and 100% were selected and excised manually using a clean razor blade. Samples were prepared for mass spectrum analysis using a slight modification of a previously described procedure [19,21]. Peptides were desalted using a C18 ZipTip (Millipore, Bedford, MA, USA), according to manufacturer’s instructions. Analyses were performed using a Bruker Daltonics Autoflex (Bruker Daltonics, Bellerica, MA, USA). The peptides mixture was analyzed using a saturated solution of α-cyano-4-hydroxycinnamic acid (HCCA) in 50% acetonitrile/0.1% trifluoroacetic acid. Lists of masses from tryptic peptides were generated and compared to the helminths databases in NCBI (http://www.ncbi.nlm.nih.gov/, accessed on 21 April 2021), using the MASCOT search program (Matrix Science, Ltd., UK) with the following parameters: one missed cleavage allowed, carbamidomethyl cysteine as the fixed modification, and oxidation of methionine as the variable modification, mass tolerance 200 ppm. Proteins with a score higher than 67 and a threshold of significance of *p* < 0.05 for extensive homology were accepted as positive. Additionally, peptide sequences were analyzed using BLASTP against the *Taenia solium* Genome BioProject PRJNA170813 (http://www.ncbi.nlm.nih.gov/bioproject/PRJNA170813, accessed on 28 April 2021 and https://parasite.wormbase.org/Taenia_solium_prjna170813/Info/Index/, accessed on 26 October 2022) databases. Classification of proteins was based on Gene Ontology Biological Process information available from the UniProtKB database (http://www.uniprot.org/, accessed on 5 June 2021).

## 3. Results

### 3.1. Immune Response in the T. crassiceps Cysticercosis Model in C57BL/6 Mouse

IL-12 and IFN-γ showed similar levels in both infected and non-infected groups at 2 and 4 weeks (Figure 1A). However, at 8 weeks, both cytokines showed a statistically significant increase in the infected group. TNF levels in the non-infected and infected group at 2 and 8 weeks were higher than the levels at week 4, although the differences were not statistically significant in the control group. Whereas, in the infected group, a significant increase in TNF production was observed in the 8-week mice. The amount of IL-6 was similar in the non-infected control group at weeks 2, 4, and 8. In the infected group, a statistically significant increase was detected at week 4, as well as a statistically significant decrease at 8 weeks. IL-5 presented a baseline level in the non-infected group.

In contrast, in infected groups, a significant increase at 2 weeks was observed, which gradually decreased its levels by half at week 8, whereas IL-4 maintained basal levels at 2 and 4 weeks in non-infected and infected groups, but its levels decreased by half at week 8 in the infected group. Figure 1B shows the average number of cysticerci increased during infection. In week 2, an average of 8.8 cysticerci, in week 4 of 15.4, and in week 8 of 79.9, were recorded.

### 3.2. Proteomics Analysis and Mass Spectrometry Protein-Identification

Figure 2 shows master gel images constructed from 2DE gels with cysticerci extracts from 2-, 4-, and 8-weeks infected mice. Differences in intensities of expression were noticeable, and the proteomics analyses detected 132 spots at 2 weeks, 137 spots at 4 weeks, and 116 spots at 8 weeks of infection. Most spots were distributed within the proteomics map in a molecular weight range of 5 to 150 kDa and a pH range of 4.0 to 9.4.

Figure 3 shows the differential expression of 15 spots randomly chosen, expressed in optical density after normalization with TPI. These spots were classified into two different groups, according to the protein differential expression. In group 1, 7 proteins with low expression at week 2 significantly increased their expression at week 4 and returned to low level at week 8. In group 2, 8 proteins were found in which expression was high at week 2, and gradually decreased at 8 weeks.

Table 1 reveals the identification of 11 of the 15 proteins of *T. crassiceps* cysticerci. Results of the MASCOT analysis showed that these proteins match those of cestodes with a sequence coverage of 19 at 58%. In addition, all sequences matched the *T. solium* genome, obtaining the contig-ubication and the ID of the gene.

In group 1, the 14-3-3 protein zeta/delta (spot-3201, KCIP 1) was identified, which is an isoform of family adapter proteins [22]. In parasites, it recognizes and binds specifically to certain sites of phosphorylated proteins [23] that are involved in cell cycle and physiology regulation, apoptosis, control of gene transcription [24,25,26], evasion mechanisms, and establishment of infection; therefore, they have been considered as antigens to develop a vaccine for *Echinococcus multilocularis* [27,28].

Myosin regulatory light chainγ (spot-3202, MRLCγ) belongs to a large family of proteins involved in the contraction of the smooth muscle [29,30]. In parasites, these proteins play a critical role for migration because vigorous muscular movement is necessary for the penetration and tissue migration. Moreover, they are considered promising vaccinal antigens because they lead to the reduction in the trematodes load, such as *Schistosoma mansoni*, or the production of eggs. When they are phosphorylated, they generate a calcium influx, leading to parasite death due to the action of eosinophils [31]. In addition, activated platelets produce myosin light chain 9 networks (Myl9 nets), which recruit CD69-expressing inflammatory cells to inflamed tissues, leading to exacerbation of inflammation [32].

Annexin B7 (spot-3306, AnxB7) belongs to a family of proteins that can bind membranes via the Ca2^+^ ion, and negatively charged phospholipids are considered scaffolding proteins that participate in membrane dynamics. AnxB7 is involved in cell motility, endocytosis, fibrinolysis, ion channel formation, cell matrix interactions, apoptosis, angiogenesis, and anticoagulation [33,34]. In the genome of *T. solium*, 13 genes coding for the annexin family have been identified. AnxB7 forms a complex with members of the endosomal sorting complex required for transport (ESCRT-III), which helps to excise and repair the damaged region of the plasma membrane, including lipid peroxidation damage [35,36].

Tubulin β 2C chain (spot 3401), β-tubulin, and α-tubulin are components of microtubules and are important in cell division and intracellular trafficking. They are related to cysticerci plasticity, motility, dynamicity, and shape [37,38,39,40].

The 60-kDa heat-shock protein (spot-3402, HSP60) is expressed by cells exposed to stress or immune activation and during inflammation. HSP60 inhibits T cell chemotaxis and stimulates the production of IL-10 in T cells, shifting the profile toward Th2. Inside the cell, HSP60 is a chaperone [41,42]. The *Schistosoma japonicum* HSP60 promotes immunosuppressive Tregs by inducing TGF-β and IL-10 production and reducing mouse liver immunopathology [43].

In group 2, proteins such as 14-3-3 epsilon protein (spot 3102) were found, which is a regulator of the TrkB/PI3K/Akt pathway [44]. The 14-3-3 epsilon protein has been identified as a molecule recruited by the TNRF2 forming a complex that inhibits NF-kB and promotes M2 response [45].

ATP synthase subunit beta (spot 3404) is involved in ATP synthesis-coupled proton transport [46]. ATP synthase subunit beta has been identified in *T. solium* and in *Trichinella britovi* as an antigen of adult worm in early infection [47,48]. It is a protein that stimulates the production of antibodies, in addition to being a signaling protein [49].

Protein disulfide isomerase (spot 4401, PDI) is a ubiquitous REDOX and multifunction enzyme, belonging to the thioredoxin family, which catalyzes dithiol–disulfide exchange reactions and displays chaperone activity. The increase in PDI is involved in pathogen infection processes and plays an important virulence role during host infection [50,51]. Specific PDI inhibitors abolished the enzymatic activity and markedly affected parasite growth [52,53,54].

Paramyosin (spot 4501, Pmy). In *T. solium*, Pmy is a component of the musculature but has also been found associated with the tegumentary cytons and is released through the tegument. *T. solium* Pmy induces significant levels of protection in cysticercosis by *T. crassiceps* and schistosomiasis inducing a Th1-like immune response [55,56,57]. Moreover, passive immunization with a Pmy monoclonal antibody reduces the parasite load. In addition, Pmy plays a role in immune evasion by interfering with complement activation binding to C1q, C8, and C9 proteins [58,59]. Therefore, it has been considered a target for the creation of vaccines for diverse helminths.

Enolase 3 (spot 5302, Eno3, 2-phospho-D-glycerate hydrolase) is a glycolytic enzyme that catalyzes the conversion of 2-phosphoglycerate to phosphoenolpyruvate, it is excreted/secreted in helminths, acting as a plasminogen-binding protein and favoring the fibrin degradation to facilitate establishment of parasites in their hosts [60,61,62,63,64,65,66]. In *T. solium*, four enolase isoforms TsEno1, TsEno2, TsEno3, and TsEno4 have been identified. The last one lacks the plasminogen-binding motif [65]. Moreover, enolase has also been used as a diagnostic and vaccine target [49].

Fasciclin-1 (spot 5506, Fas-1). Three isoforms, Fas-1, 2, and 3 exist in *T. solium* cysticerci. These proteins are constitutively expressed in cysticerci and adult stages, preferentially located in the scolex and the vesicular wall, and are biomarkers for chronic neurocysticercosis (NCC) [67,68,69]. It has been reported that Fas-1 and 2 of *T. solium* cysticerci bind to plasminogen to convert it to plasmin, inhibiting the complement pathway and favoring immune evasion [65,70,71].

Some proteins from both groups were not identified, but their differential expression was very interesting (Figure 3). In group 1, spots 3002 and 7204, whereas in group 2, were spots 3105 and 6302.

## 4. Discussion

*Taenia solium* and *T. crassiceps* cysticerci cause cysticercosis in humans, pigs, and mice. Their establishment and survival depend on the environment encountered in hosts [5,6,7]. It has been demonstrated that BALB/c mice are more susceptible to parasite infection than C57BL/6 mice [5]. In the *T. crassiceps* cysticercosis model with susceptible BALB/c mice, cysticerci products shift to a Th2 response after 4 weeks, allowing for parasite growth, whereas resistant C57BL/6 mice develop a strong Th1 response that limits parasitic growth [9]. Despite the well-documented information about the immune response in both mice strains, there is limited information on how cysticerci respond to the immune response environment in resistant mice.

In this study, we developed a Th1 environment in the *T. crassiceps* cysticerci model with resistant C57BL/6 mice, which is confirmed by the sustained increase in IL-12, INF-γ, and TNF and a low parasitic load during 8 weeks of infection. This contrasts with previous results with susceptible BALB/c mice where the Th1 response is lost at week 4 of infection and replaced by a Th2 response with exponential parasites growth [19,71,72].

This sustained Th1 response and low parasite burden are consistent with previous reports on the control of parasite growth [73,74,75]. A prolonged Th1 environment does not prevent infection but slows the ability of cysticerci to reproduce [7,14]. Likewise, proteomics analysis allowed us to determine the differential protein expression profile in *T. crassiceps* cysticerci obtained under a Th1 environment in C57BL/6 mice. We succeeded in identifying 11 proteins; their differential expression allowed for the formation of 2 groups: The expression of proteins in group 1 was upregulated at week 4 and showed a significant downregulation at weeks 2 and 8. Group 2 proteins were highly expressed at week 2 and the expression was gradually downregulated until week 8.

It is noteworthy that high paramyosin levels correlate with high IL-5 levels during the second week of infection. This is consistent with previous reports where individuals resistant to reinfection by *S. mansoni* present high levels of anti-paramyosin antibodies and high levels of IL-2 and IL-5. In addition, it is known that IL-5 and IL-6 participate in the recruitment of eosinophils, which are considered important effectors in the destruction of parasites in porcine cysticercosis [76,77].

In addition, Eno3 is differentially expressed from week 2. It has been found to be expressed in cells with Th1/Th17 immune responses during the acute phase of experimental neurocysticercosis in *T. crassiceps*, so these responses could be present in resistant mice [78,79].

Of note, both BALB/c and C57BL/6 mice express proteins involved in establishment, reparation, and defensive roles that prevent the host’s immune attack, such as inhibition of complement and maintenance of the vesicular wall structure of cysticerci, for example Pmy, Eno, Fas-1, and Anxs, which are shared in both mice strains [19,38,39,46,50,51,65,69,80,81,82].

The downregulated expression of all proteins at week 8 in C57BL/6 mice could be indicating adaptation to the Th1 environment by cysticerci. It agrees with a previous report for C57BL/6 mice, where it controls *T. crassiceps* cysticerci growth, but the infection persists, and parasites reproduce slowly [9].

On the other hand, the data presented here demonstrate that the immune response against *T. crassiceps* differs between susceptible and resistant mice. C57BL/6 mice developed a sustained and strong Th1 response up to 8 weeks. This shows that the genetic background of this mouse strain is important to mounting a sustained Th1 response. For example, the expression of a nonclassical class I major histocompatibility complex (MHC) Qa-2 antigen in C57BL/6 mice is related to *T. crassiceps* cysticercosis resistance, whereas non-expression in BALB/c mice makes them susceptible [5].

A Th2 response and parasite regulatory factors are important for the survival of both the parasite and the host, but other factors, like hormones, could also be involved, for example, testosterone and 17-beta estradiol stimulate the reproduction and infectivity of the cysticercus while prostaglandin E2 favors the Th2 response, by downregulating IL-12 and IL-2 [83,84,85].

## 5. Conclusions

The 2DE and MALDI-TOF mass spectrometry methods allowed us to determine the protein expression profile of cysticerci during an infection of C57BL/6 mice in a Th1 environment. In this environment, parasites express proteins that help their establishment, repair of their vesicular walls, and protection against the immune responses of mice. Furthermore, our data support that the genetic background of the mice strains and a sustained Th1 response during infection with *T. crassiceps* cysticerci, control cysticercosis. Finally, these proteins could be targets for the development of vaccines or drugs against helminths.

## Figures and Tables

**Figure 1 pathogens-12-00678-f001:**
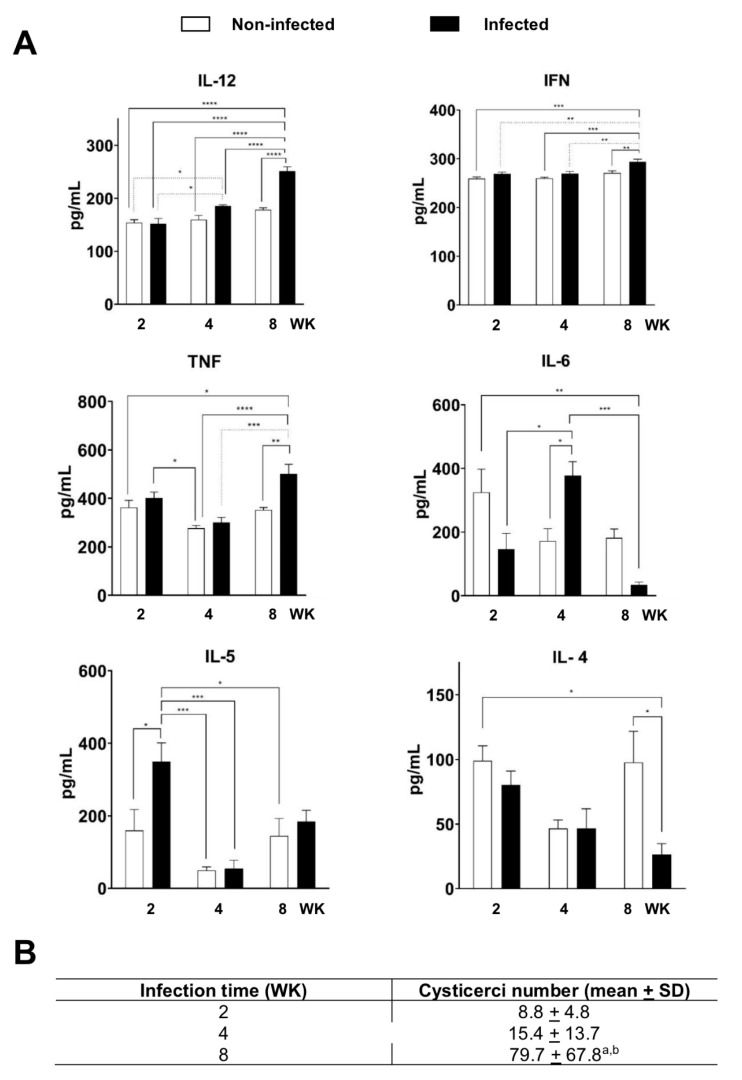
Cytokine levels in C57/BL/6 mice with cysticercosis at 2, 4, and 8 weeks (WK) after stimulation with *T. crassiceps* crude extract. (**A**) Levels of IL-12, INF-γ, TNF, IL-6, IL5, and IL-4 in the supernatants of cultured splenocytes determined with ELISA, significance of * *p* < 0.05; ** *p* < 0.01; *** *p* < 0.001; **** *p* < 0.0001. (**B**) Average number of cysticerci in infected mice at 2, 4, and 8 weeks (WK). Mean for each experiment was measured for *n* = 18 (2 WK), *n* = 8 (4 WK), and *n* = 5 (8 WK). ^a^
*p* < 0.05 vs. 2 weeks; ^b^
*p* < 0.05 vs. 4 weeks. One-way ANOVA and Tukey’s multiple comparisons test.

**Figure 2 pathogens-12-00678-f002:**
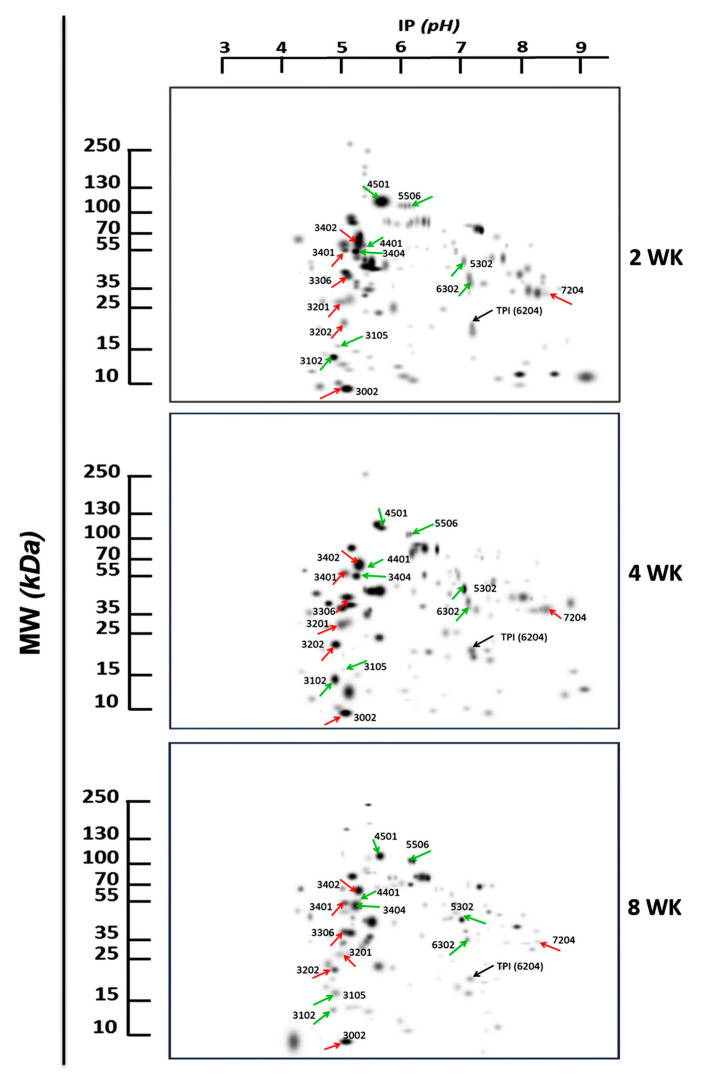
Representative 2DE proteomics profiles of *T. crassiceps* cysticerci in infected mice at 2, 4, and 8 weeks (WK). Molecular weight (MW) and isoelectric point (IP) are indicated in kDa and pH, respectively. The arrows and numbers indicate the spots of proteins with a differential expression between 70% and 100% and sequenced by MALDI-TOF MS. TPI was used as a constitutive protein (black arrow). Red arrows show group 1, and green arrows show spots of group 2.

**Figure 3 pathogens-12-00678-f003:**
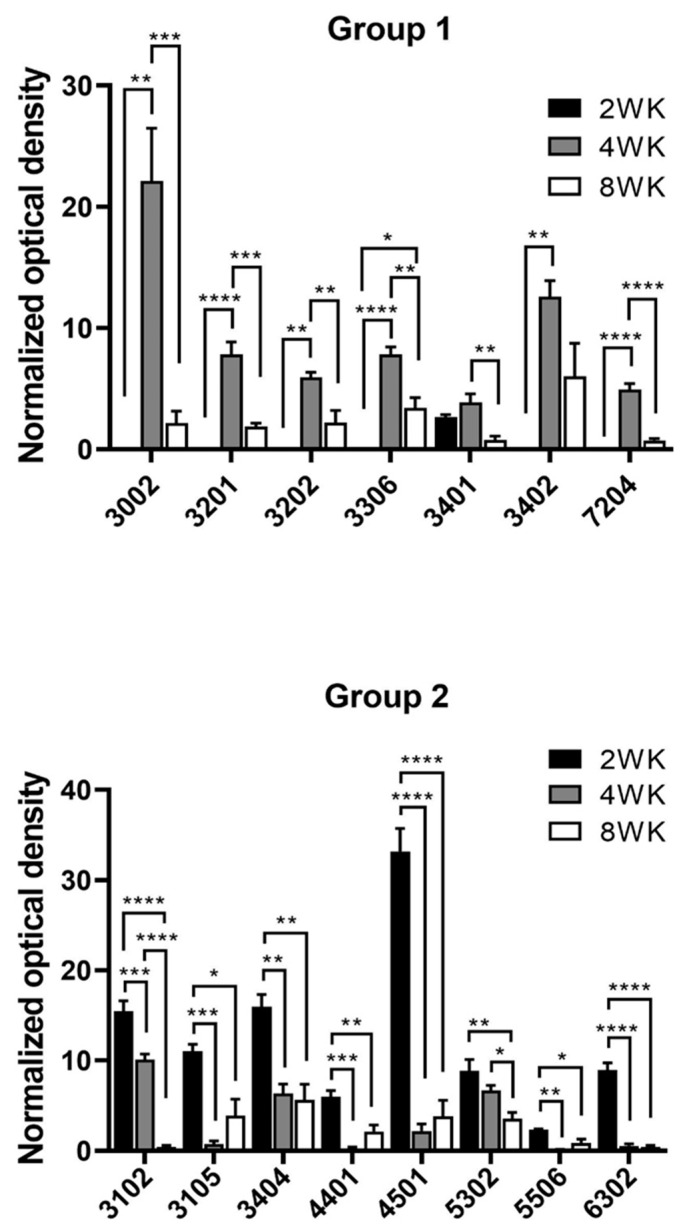
Profiles of the differential expression of proteins by *T. crassiceps* cysticerci at 2, 4, and 8 weeks (WK) post-infection. Two groups were identified: Group (1) low expression at 2 weeks that increased at the 4th week and decreased on the 8th week, and Group (2) high expression at 2 weeks that decreased at 4 and 8 weeks, the last being the lowest. Significance of * *p* < 0.05; ** *p* < 0.01; *** *p* < 0.001; **** *p* < 0.0001.

**Table 1 pathogens-12-00678-t001:** Protein identification from *Taenia crassiceps* ORF cysticerci by amino acid sequence and BLAST analysis with the Mascot and *Taenia solium* database genome. Gene Ontology program was used to identify process and molecular function. The theoretical Ip and MW were calculated using the sequence of each protein.

Protein (Spot number)	MASCOT		*Taenia solium* PRJNA170813	Theoretical data MW	Access to UniprotKB
	Score	Sequence coverage (%)	Contig/Gene ID	(kDa/pI (pH)	
GROUP 1					
14-3-3 protein zeta/delta (3201)	78	36	00616/TsM_000719200	28,069.73/4.63	U6JEE0_ECHGR
Myosin regulatory light chain y (3202)	94	49	00860/TsM_001153700	21,929.57/4.53	A0A068YDE7_ ECHMU
Annexin B7 (3306)	97	27	00183/TsM_000793200	111.55/4.39	Q52MU2 _TAESO
Tubulin beta 2C chain (3401)	112	19	01071/TsM_001072900	112.04/4.49	A0A068WIF6_ ECHGR
Heat shock protein 60 kDa protein (3402)	105	41	00260/TsM_000546800	60,568.63/5.09	A3F4T7 _TAEAS
GROUP 2					
14-3-3 protein epsilon (3102)	110	58	00306/TsM_000569000	29,217.94/4.70	U6JEE0 _ECHGR
ATP synthase subunit beta (3404)	104	34	00029/TsM_000805000	55,799.04/5.39	A0A0R3VYB8 _TAEAS
Protein disulfide isomerase (4401)	100	39	04289/TsM_000671300	55,214.79/4.61	A0A068WY47_ECHGR
Paramyosin (4501)	100	36	00569/TsM_001115200	107,772.95/5.00	Q68J63_ TAESO
Enolase 3 (5302)	121	42	02256/TsM_000447100	46,557.32/6.90	A0A2P1AM74 _TAESO
Fasciclin-1 (5506)	106	29	00312/TsM_000655200	95,014.71/6.66	X2D553_TAESO

## Data Availability

Not applicable.

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
