# Peer review of "Differential Protein Expression of Taenia crassiceps ORF Strain in the Murine Cysticercosis Model Using Resistant (C57BL/6) Mice"

_pathogens, 2023, doi:10.3390/pathogens12050678_

Round 1

Reviewer 1 Report

The manuscript "Differential protein expression of Taenia crassiceps ORF strain in the murine cysticercosis model using resistant (C57BL/6) mice" has clear scientific importance, but fails to adequately present the results to readers. I suggest:

1. Improve the introduction and description of the hypothesis and importance of the research;

2. Considerably improve the presentation of results, especially of differentially expressed proteins;

3. Improve the quality of figure 2.

4. Considerably improve the discussion of results and include a proper conclusion;

5. Revise English.

Author Response

ANSWER TO COMMENTS AND SUGGESTIONS FOR REFEREE 1.

Manuscript ID: pathogens-2332013.

Article

Differential protein expression of Taenia crassiceps ORF strain in the murine cysticercosis model using resistant (C57BL/6) mice.

Lucía Jiménez•1, Mariana Díaz-Zaragoza•*1,5, Magdalena Hernández4, Luz Navarro3, Ricardo Hernández-Ávila2, Sergio Encarnación-Guevara4, Pedro Ostoa-Saloma2 and Abraham Landa*1

Introduction

47 – 51 The authors include information regarding murine infection models with T. crassiceps, both for susceptibility (BALB/c) and resistance (C57BL/6), and some parasite molecules that could be modulating this response... However, nothing is mentioned about the works previously published by Toledo et. to the. 2001 and Vega-Angeles et. to the. 2019, where they refer to the role of theKETc-1 and KETc-2 proteins and the Glutathione transferases produced by cysticerci, which function as antigens by stimulating aTh1 response. Additionally, detoxifying enzymes such as Cu/Zn-superoxide dismutase, typical 2-Cys-peroxiredoxin, and Glutathione transferases that the cysticercus produces during the immune response are not mentioned, as previous studies have shown. It is recommended to include this information to enrich the manuscript.

Answer: We have fixed the introduction and expanded the immune response section to include the referees' suggestions. Please see lanes 50-56

Materials and Methods

65 – 69 It is mentioned that cysticerci extraction is performed with PBS; however, since these are proteomic studies, only freezing can maintain the integrity of the protein extracts? No solution that protects the proteins from degradation before their purification and/or freezing of the cysticerci is used?

Answer: The protease inhibitor used in washing cysticerci was added (see line 74-75).

73 Correct the number of cells (2 × 106/mL). Place number 6 in superscript (2 × 106/mL).

Answer: It has been corrected. Please see lane 79.

2.3. Spleen cell isolation and cytokine detection

For lymphoproliferation assays, Concanavalin or LPS are sometimes used as controls. For the stimulation with the crude extracts of T. crassiceps, some of these controls were used, or is it correct to use a positive control for these experiments?

Answer: Con A was used to check for lymphocyte proliferation, but this was not written in the article. The crude extract of cysticerci was used as a secondary stimulus to determine the specific immune response to these antigens.

Results

A legend to figure. In Figure 1B (Table), it is recommended to replace the # and $signs with the superscripts a, b .

139 Correct the symbols # and $ in the footer of the figure.

Answer: The signs have been replaced by the letters a and b both in the table of figure 1 and in the footer of the figure. Please see lane 144

148 In the text, it is recommended to place "master gel images."

Answer. It has been corrected. Please see lane 153

Figure 2

Improving the quality of the images that correspond to the 2DEgels is recommended.

Answer: The image has been improved according the recommendation of the Journal.

Discussion

1) Based on the information obtained from protein sequencing and at the functional level, it is crucial to correlate the levels of cytokines at 2, 4, and 8 weeks with the relative expression levels of the proteins that could be related to the regulation of the protective response.

Answer: It was done in the discussion section. Please see lines 269-274 and les 284-293.

2) In lines 280-290, mention is made of proteins related to establishing and evading the immune response. However, nothing is discussed about the results obtained in similar works, where they evaluate the protein profiles obtained in 2DE but in a susceptible murine model with BALB/c mice (Díaz-Zaragoza et al.2020). It would be interesting if the authors could discuss the differences or the similarities that are presented

Answer: Answer: It was done in the discussion section. Please see lines 294-297.

3) Line 285 mentions, "In contrast, BALB/c mice developed a short (2 wk) Th1 response followed by a change to a long Th2", which is understood as if these results of the characterization of the immune response in BALB/c mice, would have been performed in this work, which did not occur. It is recommended to change the wording so that it is clearly being compared with results previously obtained in other reports.

Answer: The redaction was corrected please see line 271-273.

4) For some of the experiments in this work, only female mice were used, which is interesting since previous reports evaluate the role of hormones in regulating the immune response. In this regard, it would be essential to discuss the role of hormones such as E2 in the regulation of the response by Taenia, and in the polarization and/or regulation of the Th1 and Th2 responses.

Answer: We have added the suggestion of the referees. Please lanes 311-314

5) As previously mentioned, some studies report the role of theKETc1 and KETc2 antigens in the polarization of the Th1 response. However, it is interesting that in this work, none of the spots reported in the 2DE correspond to any of these proteins. Due to their importance in the immune response, it would be necessary for the authors, based on these results, to discuss the implication of the presence or absence of these antigens in the course of the immune response generated by the T. crassiceps cysticerci.

Answer: Because the proteins were randomly selected from over 120 differentially expressed dots, it is possible that these antigens were found within the proteins that were not selected.

6) In this work, part of the results presented here refer to evaluating some cytokines related to Th1 and Th2 responses. However, in the literature, some reports address the role of theTh17 response during some pathologies caused by cysticerci (Vania et al. 2015; Peón 2016; Arora 2017). Even reports of proteomic analyses evaluate the effect of the secretion/excretion antigens of Cysticercus cellulosae on the TGF-β signaling pathways and the TH17-type response. Due to the above, and based on the results obtained, it would be interesting if the authors could retake these ideas and discuss them to enrich the work.

Answer: We have added the suggestion of the referee. Please lanes 290-293.

Generals

It is recommended to improve the wording of the results since it can become confusing in some paragraphs. Likewise, as previously mentioned, it is essential to enrich the introduction with more information on the previous reports on the cysticerci proteins involved in regulating the host's immune response. Additionally, itis crucial to improve the quality of the images of the master gels. Finally, it is considered essential to specify the ideas expressed in the discussions with a section of conclusions and future directions to complete the general information of the manuscript.

Answer: all these suggestions have been done.

At final a conclusion was added, also we correct the references section.

Reviewer 2 Report

MANUSCRIPT´S REVIEW

In this work, the authors present a protein characterization of the course of T. crassiceps infection in a murine model with C57BL mice, which has been shown to induce a Th1 response that confers resistance to infection, reducing the number of cysticerci. The results presented are an advance in knowledge in the biology of Taenia and the protein factors associated with the disease, constituting a significant advance in the area. The work is based on methodological tools widely used in proteomic studies of the infection caused by Taenia and the minimum elements necessary to know the associated immune response. However, the authors must address some observations to enrich the manuscript.

Introduction

47 – 51 The authors include information regarding murine infection models with T. crassiceps, both for susceptibility (BALB/c) and resistance (C57BL/6), and some parasite molecules that could be modulating this response... However, nothing is mentioned about the works previously published by Toledo et. to the. 2001 and Vega-Angeles et. to the. 2019, where they refer to the role of the KETc-1 and KETc-2 proteins and the Glutathione transferases produced by cysticerci, which function as antigens by stimulating a Th1 response. Additionally, detoxifying enzymes such as Cu/Zn-superoxide dismutase, typical 2-Cys-peroxiredoxin, and Glutathione transferases that the cysticercus produces during the immune response are not mentioned, as previous studies have shown. It is recommended to include this information to enrich the manuscript.

Materials and Methods

65 – 69 It is mentioned that cysticerci extraction is performed with PBS; however, since these are proteomic studies, only freezing can maintain the integrity of the protein extracts? No solution that protects the proteins from degradation before their purification and/or freezing of the cysticerci is used?

73 Correct the number of cells (2 × 106/mL). Place number 6 in superscript (2 × 106/mL).

23. Spleen cell isolation and cytokine detection

For lymphoproliferation assays, Concanavalin or LPS are sometimes used as controls. For the stimulation with the crude extracts of T. crassiceps, some of these controls were used, or is it correct to use a positive control for these experiments?

Results

A legend to figure

In Figure 1B (Table), it is recommended to replace the # and $ signs with the superscripts a,b

139 Correct the symbols # and $ in the footer of the figure.

148 In the text, it is recommended to place "master gel images."

Figure 2

Improving the quality of the images that correspond to the 2DE gels is recommended.

Discussion

It is highly recommended to enrich the discussion based on the results obtained. Therefore, it is recommended that the authors include the following points:

1) Based on the information obtained from protein sequencing and at the functional level, it is crucial to correlate the levels of cytokines at 2, 4, and 8 weeks with the relative expression levels of the proteins that could be related to the regulation of the protective response.

2) In lines 280-290, mention is made of proteins related to establishing and evading the immune response. However, nothing is discussed about the results obtained in similar works, where they evaluate the protein profiles obtained in 2DE but in a susceptible murine model with BALB/c mice (Díaz-Zaragoza et al. 2020). It would be interesting if the authors could discuss the differences or the similarities that are presented.

3) Line 285 mentions, "In contrast, BALB/c mice developed a short (2 wk) Th1 response followed by a change to a long Th2", which is understood as if these results of the characterization of the immune response in BALB/c mice, would have been performed in this work, which did not occur. It is recommended to change the wording so that it is clearly being compared with results previously obtained in other reports.

4) For some of the experiments in this work, only female mice were used, which is interesting since previous reports evaluate the role of hormones in regulating the immune response. In this regard, it would be essential to discuss the role of hormones such as E2 in the regulation of the response by Taenia, and in the polarization and/or regulation of the Th1 and Th2 responses.

5) As previously mentioned, some studies report the role of the KETc1 and KETc2 antigens in the polarization of the Th1 response. However, it is interesting that in this work, none of the spots reported in the 2DE correspond to any of these proteins. Due to their importance in the immune response, it would be necessary for the authors, based on these results, to discuss the implication of the presence or absence of these antigens in the course of the immune response generated by the T. crassiceps cysticerci.

6) In this work, part of the results presented here refer to evaluating some cytokines related to Th1 and Th2 responses. However, in the literature, some reports address the role of the Th17 response during some pathologies caused by cysticerci (Vania et al. 2015; Peón 2016; Arora 2017). Even reports of proteomic analyses evaluate the effect of the secretion/excretion antigens of Cysticercus cellulosae on the TGF-β signaling pathways and the TH17-type response. Due to the above, and based on the results obtained, it would be interesting if the authors could retake these ideas and discuss them to enrich the work.

Generals

It is recommended to improve the wording of the results since it can become confusing in some paragraphs. Likewise, as previously mentioned, it is essential to enrich the introduction with more information on the previous reports on the cysticerci proteins involved in regulating the host's immune response. Additionally, it is crucial to improve the quality of the images of the master gels. Finally, it is considered essential to specify the ideas expressed in the discussions with a section of conclusions and future directions to complete the general information of the manuscript.

Author Response

ANSWER TO COMMENTS AND SUGGESTIONS FOR REFEREE 2.

Manuscript ID: pathogens-2332013.

Article

Differential protein expression of Taenia crassiceps ORF strain in the murine cysticercosis model using resistant (C57BL/6) mice.

Lucía Jiménez•1, Mariana Díaz-Zaragoza•*1,5, Magdalena Hernández4, Luz Navarro3, Ricardo Hernández-Ávila2, Sergio Encarnación-Guevara4, Pedro Ostoa-Saloma2 and Abraham Landa*1

Introduction

47 – 51 The authors include information regarding murine infection models with T. crassiceps, both for susceptibility (BALB/c) and resistance (C57BL/6), and some parasite molecules that could be modulating this response... However, nothing is mentioned about the works previously published by Toledo et. to the. 2001 and Vega-Angeles et. to the. 2019, where they refer to the role of theKETc-1 and KETc-2 proteins and the Glutathione transferases produced by cysticerci, which function as antigens by stimulating aTh1 response. Additionally, detoxifying enzymes such as Cu/Zn-superoxide dismutase, typical 2-Cys-peroxiredoxin, and Glutathione transferases that the cysticercus produces during the immune response are not mentioned, as previous studies have shown. It is recommended to include this information to enrich the manuscript.

Answer: We have fixed the introduction and expanded the immune response section to include the referees' suggestions. Please see lanes 50-56

Materials and Methods

65 – 69 It is mentioned that cysticerci extraction is performed with PBS; however, since these are proteomic studies, only freezing can maintain the integrity of the protein extracts? No solution that protects the proteins from degradation before their purification and/or freezing of the cysticerci is used?

Answer: The protease inhibitor used in washing cysticerci was added (see line 74-75).

73 Correct the number of cells (2 × 106/mL). Place number 6 in superscript (2 × 106/mL).

Answer: It has been corrected. Please see lane 79.

2.3. Spleen cell isolation and cytokine detection

For lymphoproliferation assays, Concanavalin or LPS are sometimes used as controls. For the stimulation with the crude extracts of T. crassiceps, some of these controls were used, or is it correct to use a positive control for these experiments?

Answer: Con A was used to check for lymphocyte proliferation, but this was not written in the article. The crude extract of cysticerci was used as a secondary stimulus to determine the specific immune response to these antigens.

Results

A legend to figure. In Figure 1B (Table), it is recommended to replace the # and $signs with the superscripts a, b .

139 Correct the symbols # and $ in the footer of the figure.

Answer: The signs have been replaced by the letters a and b both in the table of figure 1 and in the footer of the figure. Please see lane 144

148 In the text, it is recommended to place "master gel images."

Answer. It has been corrected. Please see lane 153

Figure 2

Improving the quality of the images that correspond to the 2DEgels is recommended.

Answer: The image has been improved according the recommendation of the Journal.

Discussion

1) Based on the information obtained from protein sequencing and at the functional level, it is crucial to correlate the levels of cytokines at 2, 4, and 8 weeks with the relative expression levels of the proteins that could be related to the regulation of the protective response.

Answer: It was done in the discussion section. Please see lines 269-274 and les 284-293.

2) In lines 280-290, mention is made of proteins related to establishing and evading the immune response. However, nothing is discussed about the results obtained in similar works, where they evaluate the protein profiles obtained in 2DE but in a susceptible murine model with BALB/c mice (Díaz-Zaragoza et al.2020). It would be interesting if the authors could discuss the differences or the similarities that are presented

Answer: Answer: It was done in the discussion section. Please see lines 294-297.

3) Line 285 mentions, "In contrast, BALB/c mice developed a short (2 wk) Th1 response followed by a change to a long Th2", which is understood as if these results of the characterization of the immune response in BALB/c mice, would have been performed in this work, which did not occur. It is recommended to change the wording so that it is clearly being compared with results previously obtained in other reports.

Answer: The redaction was corrected please see line 271-273.

4) For some of the experiments in this work, only female mice were used, which is interesting since previous reports evaluate the role of hormones in regulating the immune response. In this regard, it would be essential to discuss the role of hormones such as E2 in the regulation of the response by Taenia, and in the polarization and/or regulation of the Th1 and Th2 responses.

Answer: We have added the suggestion of the referees. Please lanes 311-314

5) As previously mentioned, some studies report the role of theKETc1 and KETc2 antigens in the polarization of the Th1 response. However, it is interesting that in this work, none of the spots reported in the 2DE correspond to any of these proteins. Due to their importance in the immune response, it would be necessary for the authors, based on these results, to discuss the implication of the presence or absence of these antigens in the course of the immune response generated by the T. crassiceps cysticerci.

Answer: Because the proteins were randomly selected from over 120 differentially expressed dots, it is possible that these antigens were found within the proteins that were not selected.

6) In this work, part of the results presented here refer to evaluating some cytokines related to Th1 and Th2 responses. However, in the literature, some reports address the role of theTh17 response during some pathologies caused by cysticerci (Vania et al. 2015; Peón 2016; Arora 2017). Even reports of proteomic analyses evaluate the effect of the secretion/excretion antigens of Cysticercus cellulosae on the TGF-β signaling pathways and the TH17-type response. Due to the above, and based on the results obtained, it would be interesting if the authors could retake these ideas and discuss them to enrich the work.

Answer: We have added the suggestion of the referee. Please lanes 290-293.

Generals

It is recommended to improve the wording of the results since it can become confusing in some paragraphs. Likewise, as previously mentioned, it is essential to enrich the introduction with more information on the previous reports on the cysticerci proteins involved in regulating the host's immune response. Additionally, itis crucial to improve the quality of the images of the master gels. Finally, it is considered essential to specify the ideas expressed in the discussions with a section of conclusions and future directions to complete the general information of the manuscript.

Answer: all these suggestions have been done.

At final a conclusion was added, also we correct the references section.